# Trends in disease incidence and survival and their effect on mortality in Scotland: nationwide cohort study of linked hospital admission and death records 2001–2016

Paul R H J Timmers [iD],[1] Joannes J Kerssens,[2] Jon Minton,[3] Ian Grant,[2] James F Wilson,[1,4] Harry Campbell,[1] Colin M Fischbacher,[2] Peter K Joshi [iD] [1]

¹Centre for Global Health Research, The University of Edinburgh Usher Institute, Edinburgh, UK
²Information Services Division, NHS National Services Scotland, Edinburgh, UK
³Public Health Observatory, NHS Health Scotland, Glasgow, UK
⁴MRC Human Genetics Unit, The University of Edinburgh MRC Institute of Genetics and Molecular Medicine, Edinburgh, UK

**Correspondence to**
Mr Paul R H J Timmers;
paul.timmers@ed.ac.uk

## ABSTRACT

**Objectives** Identify causes and future trends underpinning Scottish mortality improvements and quantify the relative contributions of disease incidence and survival.

**Design** Population-based study.

**Setting** Linked secondary care and mortality records across Scotland.

**Participants** 1 967 130 individuals born between 1905 and 1965 and resident in Scotland from 2001 to 2016.

**Main outcome measures** Hospital admission rates and survival within 5 years postadmission for 28 diseases, stratified by sex and socioeconomic status.

**Results** 'Influenza and pneumonia', 'Symptoms and signs involving circulatory and respiratory systems' and 'Malignant neoplasm of respiratory and intrathoracic organs' were the hospital diagnosis groupings associated with most excess deaths, being both common and linked to high postadmission mortality. Using disease trends, we modelled a mean mortality HR of 0.737 (95% CI 0.730 to 0.745) from one decade of birth to the next, equivalent to a life extension of ~3 years per decade. This improvement was 61% (30%–93%) accounted for by improved disease survival after hospitalisation (principally cancer) with the remainder accounted for by lowered hospitalisation incidence (principally heart disease and cancer). In contrast, deteriorations in infectious disease incidence and survival increased mortality by 9% (~3.3 months per decade). Disease-driven mortality improvements were slightly greater for men than women (due to greater falls in disease incidence), and generally similar across socioeconomic deciles. We project mortality improvements will continue over the next decade but slow by 21% because much progress in disease survival has already been achieved.

**Conclusion** Morbidity improvements broadly explain observed mortality improvements, with progress on prevention and treatment of heart disease and cancer contributing the most. The male–female health gaps are closing, but those between socioeconomic groups are not. Slowing improvements in morbidity may explain recent stalling in improvements of UK period life expectancies. However, these could be offset if we accelerate improvements in the diseases accounting

### Strengths and limitations of this study

► The individual-level linkage of hospital and death records in this population-wide dataset allows for direct modelling of improvements in 28 disease categories in terms of improvements in disease incidence and subsequent survival, stratified by sex and socioeconomic status.

► Exclusion of migrating individuals means changes in disease are unaffected by population shifts, and allow for diseases to be compared with each other and summarised into trends in mortality based on morbidity.

► Hospital admission diagnosis and subsequent survival avoid issues with cause of death recording; however, they do not provide evidence of the causal effect of disease on mortality and may in some cases track changes in underlying frailty.

► This study is limited to the assessment of diseases which result in a hospital admission prior to death.

for most deaths and counteract recent deteriorations in infectious disease.

## INTRODUCTION

In recent decades, there has been a substantial improvement in life expectancies at birth in the UK.[1] More recently, several studies have suggested that there has been slowdown in improvements in the USA, UK, France, Germany, Sweden, the Netherlands and other Organisation for Economic Cooperation and Development countries; however, the causes are less clear, with speculation that they may arise from slowing improvements in cardiovascular disease, increased influenza mortality and/or pressure on health and social care services.[1–8] Understanding trends in disease incidence and subsequent survival could illuminate such trends in mortality,



and disentangling how and how much different diseases contribute has the potential to reveal whether investment in healthcare and research is directed at the most urgent diseases and most affected individuals.

Through its electronic Data Research and Innovation Service (eDRIS), Scotland has linkage of historical individual death and electronic health records in a controlled environment, with specific study approvals by the Public Benefit and Privacy Panel. This allows direct modelling at an individual level of the incidence of disease and subsequent death or survival of subjects. Furthermore, because historic records are available and the whole population is covered, a retrospective cohort study can be constructed (with inherent representativeness of the initial sample, with very complete levels of follow-up, and without survivor bias).

Here, we use population-wide data between 2001 and 2016 on residents of Scotland born before 1966 to explore how trends in longevity were driven by different trends in broad classes of disease incidence or survival, and highlight diseases which have shown more or less improvement in their contribution to overall mortality. We partition overall mortality by sex and socioeconomic status and, assuming past disease improvements continue to the same extent in the future, use these results to project future improvements in mortality and their changing sources.

## METHODS

All methods and results are reported in line with REporting of studies Conducted using Observational Routinely-collected Data (RECORD) guidelines.[9]

### Data sources

We received ethical approval to access administration and care records from NHS National Services Scotland (NSS) from 2001 to 2016. The final study population included all 1 967 130 individuals born between 1905 and 1965 who registered with the NSS, were resident in Scotland during the study period and had complete and reliable records on their date of birth, socioeconomic status and death (if applicable). Linkage and quality control of the data are described below.

### Community Health Index dataset

Records were extracted from the historical and current Community Health Index (CHI) dataset. This is a register of all patients in NHS Scotland and is fed by eight regional databases (eg, GP database, cancer screening). The register is considered complete from 2001 onwards. The CHI number, contained in the dataset, is effectively a patient identifier and added to other health datasets to make linkage possible, for instance between hospital admissions, death records and the Scottish cancer registry.[10] Our extract consisted of 2 691 304 deidentified records, constituting the identified population of Scotland in 2001 who had been born between 1905 and 1965.

The Scottish Index of Multiple Deprivation (SIMD)[11] was used to quantify socioeconomic status, determined by individuals' full postcode, and subsequently converted into deciles. The dataset we received contained only records with district-level postcodes and SIMD deciles, of which we excluded individuals with district codes with less than 5000 individuals (thereby excluding anomalous postcodes, often with special meanings, such as 'marketing campaign'; n=11 564). We also excluded individuals missing from the CHI database in 2016, but not recorded as dead (and therefore likely transferred out of Scotland; n=573 711), individuals with record discrepancies between the CHI and National Records of Scotland databases (n=79 131) and individuals with records outside of the study dates or missing information on socioeconomic class (n=59 767), giving 1 967 130 individuals for analysis after quality control (online supplementary file 1). Characteristics of the excluded individuals were similar to the rest of the population, except for postcode exclusions and database transfers, which were missing socioeconomic information and death records, respectively, as expected (online supplementary file 2).

### National Registry of Scotland death records

We received 1 477 796 death records from the National Registry of Scotland (NRS) of all deaths occurring between 1990 and 2016, of which 699 093 could be matched to the CHI database before quality control. Unmatched records were usually for deaths occurring prior to the study start (2001). Of the matched records, 176 197 belonged to individuals who were excluded during CHI quality control, leaving 602 506 total deaths for analysis (table 1).

### Acute hospital admission

Health records were also linked to 30 054 191 acute hospital admissions, of which 17 264 379 were dated between 2001 and 2016 and could be matched (online supplementary files 3 and 4).

### Disease classification

The main diagnosis of acute hospital admission records, excluding any secondary diagnoses, was used to classify records into disease categories, which corresponded to disease blocks as described in the chapters of the International Classification of Disease Codes, Tenth Revision (ICD-10).[12] In order to model the effect of disease incidence and avoid double counting of chronic conditions, we used only the first admission of a disease category for each individual, excluding subsequent visits to the hospital for diseases within the same category. The term 'incidence' is used throughout this study to refer to the first recorded hospital admission of any disease within the disease category during the study period.

### Design

Mortality trends were modelled using morbidity trends: we first determined the major disease categories (ICD-10 blocks) associated with the most lives lost by taking into account the frequency of the disease (as measured by

**Table 1** Description of the data

| Sex | Dead | N | | | Age entry | | | Age exit | | | Hospital visits | | |
|---|---|---|---|---|---|---|---|---|---|---|---|---|---|
| | | Individuals | Admitted | Hospital visits | Mean | Median | SD | Mean | Median | SD | Mean | Median | SD |
| Male | False | 633953 | 429659 | 2315915 | 50.9 | 49.4 | 10.3 | 65.6 | 64.3 | 9.8 | 3.7 | 2.0 | 6.8 |
| Male | True | 283835 | 283835 | 2742686 | 68.0 | 69.3 | 11.7 | 75.6 | 77.1 | 11.4 | 9.7 | 7.0 | 11.4 |
| Female | False | 730671 | 511632 | 2764040 | 52.7 | 51.1 | 11.6 | 67.4 | 66.1 | 10.8 | 3.8 | 2.0 | 7.0 |
| Female | True | 318671 | 318671 | 2895443 | 72.1 | 73.8 | 12.0 | 79.7 | 81.7 | 11.5 | 9.1 | 6.0 | 11.0 |
| Both | False | 1364624 | 941291 | 5079955 | 51.9 | 50.3 | 11.1 | 66.6 | 65.3 | 10.4 | 3.7 | 2.0 | 6.9 |
| Both | True | 602506 | 602506 | 5638129 | 70.1 | 71.7 | 12.0 | 77.8 | 79.5 | 11.6 | 9.4 | 6.0 | 11.2 |
| Both | All | 1967130 | 1543797 | 10718084 | 57.5 | 55.4 | 14.1 | 70.0 | 69.2 | 11.9 | 5.4 | 3.0 | 8.8 |

The population included almost 2 million individuals (one-third of whom died during the study). See online supplementary file 3 for descriptives by deprivation including ICD-10 codes.

N refers to the total number in the study. (Individuals), the population under study. (Admitted), those admitted to hospital at least once. (Hospital visits), the total number of hospital admission records ; Age entry refers to age at the start of the study period (1 December 2000); Age exit refers to age at the end of the study period (31 January 2016) or at the end of life; Hospital visits refers to the number of records of visiting the hospital per individual;
ICD-10, International Classification of Disease Codes, Tenth Revision.

hospitalisation) and its effect on survival (as measured by the subsequent all-cause mortality of patients admitted for the disease compared with the mortality of everyone else). The effect of disease incidence has previously been modelled based on 1-year, 5-year or 10-year mortality[13] ; we chose 5 years as this captured the great part of excess mortality attributable to the incidence, rather than common underlying factors, although this does vary by disease (area under graphs in online supplementary file 5, in excess of asymptotic rates) while leaving a range of 10 years in our study to examine trends over time. We combined disease frequency and 5-year excess age-adjusted death rates to calculate a burden of death weighting for each disease block. We then looked at how the age-adjusted trends in hospitalisation rates (as a proxy for incidence) changed for each disease, by decade of birth, projecting that if incidence of a disease fell by a given percentage, its contribution to mortality would fall similarly. The use of a cohort model for the incidence of disease was driven by empirical investigation. Specifically, the distinctions we found by decade of birth in cancers, especially 'Malignant neoplasm of respiratory and intra-thoracic organs' (C30–C39) in online supplementary files 6 and 7, show a clear cohort effect. However, it should be recognised that the cohorts have only been observed over the study period (2001–2016). After calculating hospitalisation rates between decades of birth, we calculated their weighted average, reflecting the expected effect of all measured disease incidence changes on mortality rates, driven by decade of birth. Similarly, we looked at how the (age-adjusted) 5-year survival rates following first hospitalisation changed by year of hospitalisation. For each block this again gives a contribution towards reduced mortality, and the weighted average, the expected effect of changes in survival of the combined diseases on overall mortality. Adding these effects (and noting we assessed changes in survival from incidences over one decade) gives the expected effect on overall mortality from decade of birth to subsequent decade of birth from the effect of changes in disease incidence and survival, under the (necessarily simplified) model that incidence is a function of birth cohort and survival post incidence is a function of year of incidence.

## Statistical analysis
### Mortality
A Cox proportional hazards model using NRS mortality data—fitting sex, decade of birth and deprivation—was used to quantify mortality in the Scottish population during the study period. The same analysis was run stratified by sex and deprivation. Unless otherwise stated (eg, median age differences in Kaplan-Meier curves) years of life of a hazard effect have been calculated by multiplying the $\log_e$ hazard ratio (lnHR) by 10 (ref. [14]). Only individuals with complete records were included in the analysis.

## Morbidity

We grouped the main diagnoses of each NHS hospital admission into categories, as laid out by the ICD-10 chapters, and included only the first instance of admission for a category per individual (discarding subsequent repeat visits to hospital for a disease within the same disease category). Analysis was restricted to more common disease blocks. Visual inspection suggested a pragmatic threshold of at least 15 000 first-time admissions (see online supplementary file 8 for all disease categories meeting this threshold).

Effects on the incidence of hospitalisation for the more common disease blocks was quantified using Cox proportional hazard models based on age, with events defined as the first incidence of hospitalisation. We fitted sex, deprivation and decade of birth as covariates. Again, the same analyses were performed stratifying by sex and deprivation.

In order to quantify all-cause mortality in the 5 years following hospitalisation, person-time of individuals was divided into phases, corresponding to the study start until hospitalisation, the first 5 years after hospitalisation and the remaining time in the study. For example, an individual admitted to hospital in 2004 for ischaemic heart disease I20–I25 (IHD) and surviving until 2010 would contribute three phases to the model: one for the period until hospitalisation (no event), one for the first 5 years after hospitalisation (no event) and one >5 years after hospitalisation (event after 1 year). The status of each phase was fitted as a covariate in a Cox proportional hazards model[15] with death as the event, adjusting for sex and deprivation:

$$h\left(x\right) = h_0\left(x\right) e^{\beta_1 X_1 + \beta_2 X_2 + \beta_3 X_3 + \beta_4 X_4} \quad (1)$$

where $h_0$ is the baseline hazard, x the patient age and $X_1$–$X_4$ the covariates sex, deprivation and logically coded phase status (0–5 years true/false and >5 years true/false), with corresponding effect sizes $\beta_1$–$\beta_4$. This yielded estimates of the proportional hazard of status (0–5 and >5 years) after hospitalisation compared with prehospitalisation mortality. Thus, the baseline hazard is a function of age and the hazard ratios reflected the effects of the other covariates. The same model was run, stratified by sex and deprivation.

## Burden

For disease blocks with at least 15 000 first admissions during the study period, the relative mortality burden of each disease block was calculated as the excess mortality in the 5 years after hospital admission (Equation 1) multiplied by the number of first-time admissions for the disease block, as follows:

$$N_{firstadmission} / N_{total} * h_{(0,5)} \quad (2)$$

where $N_{firstadmission}$ is the total number of first hospital admissions of the disease category during the study period, $N_{total}$ is the total number of individuals in the study and $h_{(0,5)}$ is the mortality of individuals in the first

5 years following hospitalisation compared with individuals who were never hospitalised for the disease category, measured in $\log_e$ hazard ratios. The resulting value was then scaled to total 1 and provides a relative measure of the number of lives lost due to the diseases within the category, with higher values indicating a disease category with more common diseases or diseases associated with higher subsequent mortality, and lower values indicating a disease category with rare diseases or diseases associated with lower subsequent mortality. While this measure may in principle be affected by differing age patterns on incidence, it was judged sufficient for our purpose—to establish broad relative weightings of the importance of each disease category.

To maintain a feasible computational burden within the national safe haven, subsequent analysis was restricted to the 25 blocks with the highest burden of death on the population (table 2). We added C50–C50 malignant neoplasm of the breast, C60–C63 malignant neoplasms of male genital organs and G30–G32 other diseases of the nervous system to this list, out of specific interest: in the sex-specific effects and awareness of the limitations of our method for Alzheimer's disease (see Discussion section). All further analyses were performed on these top 28 blocks (T-28). The use of (first) hospitalisation for a disease as our definition of incidence is imperfect (eg, for Alzheimer's disease where hospitalisation following incidence is rare or delayed, and even first diagnosis in the community will often be preceded by a long latent period).[16]

## Disease survival

Improvements in disease outcomes by ICD-10 block were calculated by comparing 5-year all-cause mortality (Equation 1) following hospitalisation in 2001 with 5-year all-cause mortality following hospitalisation in 2011. As 5-year mortality estimates in 2011 had more uncertainty (due to fewer deaths in 2011–2016), we also calculated 5-year mortality following first-time hospitalisation for every year between 2001 and 2011 (ie, mortality of patients admitted in those years), and used the trend in mortality over time to inform the 2011 estimate. To do so, we regressed the yearly mortality estimates against year of hospital admission, fitting a third-order polynomial to allow for nonlinear relationships, and weighted the estimates by the inverse of their variances to account for uncertainty:

$$y = \beta_1 x + \beta_2 x^2 + \beta_3 x^3 + \epsilon \quad (3)$$

where y is the 5-year mortality hazard after hospital admission in year x, and $\beta_1$, $\beta_2$, $\beta_3$ are the coefficients describing the relationship between y and x. We then used the value and SE predicted for 2011 by the model as our estimate for 5-year all-cause mortality for hospitalisation in 2011.

## Mortality estimates from morbidity

Estimates of the improvement in incidence of hospitalisation between decades of birth were combined into an

**Table 2** Relative mortality burden of hospital admission by disease grouping and improvements in hospitalisation incidence and survival

| ICD-10 | Disease grouping | Disease importance | | | Average 10-year improvements HR (95% CI) | | | Survival to incidence ratio (SE) |
|---|---|---|---|---|---|---|---|---|
| | | Total hospital visits | 5-Year mortality (HR) | Relative weight | Incidence | Survival | Combined | |
| J09–J18 | Influenza and pneumonia | 110 985 | 5.28 | 0.068 | 1.19 (1.11 to 1.27) | 0.86 (0.80 to 0.92) | 1.02 (0.92 to 1.12) | 0.47 (0.13) |
| R00–R09 | Symptoms and signs involving the circulatory and respiratory systems | 225 504 | 2.08 | 0.061 | 0.88 (0.85 to 0.92) | 0.79 (0.73 to 0.86) | 0.70 (0.64 to 0.76) | 0.65 (0.14) |
| C30–C39 | Malignant neoplasm of respiratory and intrathoracic organs | 54 178 | 21.12 | 0.061 | 0.83 (0.75 to 0.91) | 0.81 (0.74 to 0.89) | 0.67 (0.59 to 0.77) | 0.52 (0.15) |
| R10–R19 | Symptoms and signs involving the digestive system and abdomen | 174 055 | 2.56 | 0.060 | 0.87 (0.83 to 0.91) | 0.89 (0.82 to 0.97) | 0.77 (0.70 to 0.85) | 0.45 (0.19) |
| R50–R69 | General symptoms and signs | 157 357 | 2.67 | 0.057 | 0.90 (0.85 to 0.94) | 0.97 (0.93 to 1.02) | 0.87 (0.81 to 0.94) | 0.19 (0.19) |
| C15–C26 | Malignant neoplasms of digestive organs | 71 981 | 8.13 | 0.056 | 0.79 (0.73 to 0.85) | 0.67 (0.63 to 0.72) | 0.53 (0.48 to 0.59) | 0.63 (0.08) |
| I30–I52 | Other forms of heart disease | 142 898 | 2.70 | 0.052 | 0.78 (0.74 to 0.83) | 0.84 (0.80 to 0.87) | 0.66 (0.61 to 0.70) | 0.43 (0.06) |
| C76–C80 | Malignant neoplasms of ill defined, secondary and unspecified sites | 39 339 | 26.13 | 0.047 | 0.84 (0.76 to 0.93) | 0.69 (0.61 to 0.79) | 0.58 (0.49 to 0.69) | 0.68 (0.16) |
| Z40–Z54 | Persons encountering health services for specific procedures and healthcare | 157 841 | 2.15 | 0.044 | 1.00 (0.95 to 1.05) | 0.74 (0.69 to 0.80) | 0.74 (0.68 to 0.81) | 1.00 (0.19) |
| I60–I69 | Cerebrovascular diseases | 100 907 | 3.06 | 0.042 | 0.79 (0.74 to 0.85) | 0.82 (0.76 to 0.87) | 0.65 (0.59 to 0.71) | 0.46 (0.09) |
| K55–K63 | Other diseases of intestines | 206 178 | 1.72 | 0.041 | 0.94 (0.91 to 0.98) | 0.80 (0.77 to 0.84) | 0.76 (0.71 to 0.80) | 0.80 (0.12) |
| J40–J47 | Chronic lower respiratory diseases | 78 467 | 3.99 | 0.040 | 0.78 (0.73 to 0.84) | 0.79 (0.72 to 0.86) | 0.62 (0.55 to 0.69) | 0.49 (0.11) |
| I20–I25 | Ischaemic heart diseases | 175 605 | 1.83 | 0.039 | 0.65 (0.63 to 0.68) | 0.77 (0.72 to 0.82) | 0.50 (0.46 to 0.55) | 0.38 (0.05) |
| N30–N39 | Other diseases of the urinary system | 126 329 | 2.31 | 0.039 | 1.04 (0.99 to 1.10) | 1.24 (1.15 to 1.34) | 1.29 (1.17 to 1.42) | 0.84 (0.22) |
| K20–K31 | Diseases of oesophagus, stomach and duodenum | 172 206 | 1.83 | 0.038 | 0.72 (0.69 to 0.75) | 0.88 (0.79 to 0.96) | 0.63 (0.56 to 0.70) | 0.29 (0.11) |
| J20–J22 | Other acute lower respiratory infections | 77 520 | 3.57 | 0.036 | 1.11 (1.03 to 1.20) | 1.11 (1.06 to 1.17) | 1.24 (1.13 to 1.36) | 0.50 (0.15) |
| S70–S79 | Injuries to the hip and thigh | 78 231 | 2.64 | 0.028 | 0.87 (0.78 to 0.96) | 1.02 (0.98 to 1.06) | 0.89 (0.79 to 0.99) | 0.12 (0.14) |
| A30–A49 | Other bacterial diseases | 33 613 | 6.60 | 0.023 | 1.56 (1.39 to 1.74) | 0.94 (0.77 to 1.15) | 1.46 (1.16 to 1.84) | 0.13 (0.21) |
| T80–T88 | Complications of surgical and medical care, not elsewhere classified | 75 217 | 2.32 | 0.023 | 1.04 (0.97 to 1.11) | 0.83 (0.74 to 0.94) | 0.86 (0.75 to 0.99) | 0.84 (0.40) |
| N17–N19 | Renal failure | 37 213 | 5.14 | 0.022 | 1.02 (0.91 to 1.15) | 0.82 (0.73 to 0.91) | 0.83 (0.71 to 0.98) | 0.90 (0.42) |
| I80–I89 | Diseases of veins, lymphatic vessels and lymph nodes, not elsewhere classified | 84 073 | 1.94 | 0.021 | 0.80 (0.75 to 0.85) | 0.79 (0.72 to 0.86) | 0.63 (0.56 to 0.71) | 0.52 (0.12) |
| K90–K93 | Other diseases of the digestive system | 47 091 | 2.98 | 0.019 | 0.98 (0.90 to 1.07) | 0.83 (0.72 to 0.95) | 0.82 (0.69 to 0.96) | 0.91 (0.50) |

Continued

**Table 2** Continued

| ICD-10 | Disease grouping | Disease importance | | | Average 10-year improvements HR (95% CI) | | | Survival to incidence ratio (SE) |
|--------|------------------|-------------------|--|--|------------------------------------------|--|--|----------------------------------|
| | | Total hospital visits | 5-Year mortality (HR) | Relative weight | Incidence | Survival | Combined | |
| I70–I79 | Diseases of arteries, arterioles and capillaries | 47410 | 2.95 | 0.019 | 0.67 (0.61 to 0.74) | 0.91 (0.85 to 0.99) | 0.61 (0.54 to 0.69) | 0.19 (0.08) |
| K50–K52 | Non infective enteritis and colitis | 59183 | 2.27 | 0.018 | 0.73 (0.68 to 0.79) | 0.89 (0.84 to 0.95) | 0.65 (0.59 to 0.72) | 0.27 (0.08) |
| S00–S09 | Injuries to the head | 64925 | 2.09 | 0.018 | 0.95 (0.88 to 1.02) | 0.98 (0.90 to 1.08) | 0.93 (0.82 to 1.05) | 0.22 (0.72) |
| C50–C50 | Malignant neoplasm of breast | 39358 | 3.21 | 0.017 | 0.81 (0.74 to 0.89) | 0.31 (0.27 to 0.36) | 0.25 (0.21 to 0.30) | 0.85 (0.07) |
| C60–C63 | Malignant neoplasms of male genital organs | 22312 | 3.23 | 0.010 | 0.91 (0.79 to 1.05) | 0.50 (0.44 to 0.57) | 0.45 (0.37 to 0.55) | 0.88 (0.14) |
| G30–G32 | Other degenerative diseases of the central nervous system | 4655 | 3.29 | 0.002 | 0.75 (0.51 to 1.10) | 0.96 (0.73 to 1.28) | 0.78 (0.49 to 1.24) | 0.11 (0.44) |
| Total | | 2664631 | 2.77 | 1.000 | 0.89 (0.88 to 0.90) | 0.83 (0.76 to 0.90) | 0.74 (0.72 to 0.75) | 0.61 (0.16) |

ICD-10: diseases contained within the disease grouping, coded by ICD-10. See online supplementary file 4 for counts of 3-letter ICD-10 records within each ICD-10 block. Total hospital visits: number of first-time admissions with main diagnosis falling within the disease block. 5-year mortality: mortality within the first 5 years after admission compared with individuals who had not yet or ever been admitted for the disease group. Relative weight: relative burden of death as a function of hospital admissions and 5-year mortality, scaled to [0–1]. Incidence: average HR of being admitted to hospital for each subsequent decade of birth. Survival: all-cause mortality HR after being admitted for the disease in 2011 compared with 2001. Combined: linear combination of changes in disease incidence and survival. 95% CIs are listed in parentheses. Ratio: the ratio of absolute changes in disease survival to incidence of hospital admission, measured on log HR scale. SE is listed in parentheses. See online supplementary file 12 for these data by sex and deprivation. See online supplementary file 8 for the relative burdens of all disease groupings with more than 15000 first-time hospital admissions.
ICD-10, International Classification of Disease Codes, Tenth Revision.

overall improvement by performing a weighted sum of all diseases, with weights derived from the relative burden of death of each disease (see above).

These 10-year improvements (I) due to incidence were added to the 10-year improvements due to postincidence survival (S) to give a total improvement due to all morbidities, and proportions due to incidence/survival were calculated as the S or I/(I+S).

## Patient and public involvement
Patients and the public were not involved in the study or its design, beyond their contribution of health records. Due to the retrospective study design and anonymised nature of the records, it was not feasible to contact individual patients nor involve them in the dissemination of results.

## Summary of outcomes
### Mortality improvements
Age-adjusted falling mortality rates observed directly from NRS death records.

### Disease burden of death
Prevalence of a disease category (total number of individuals admitted at least once 2001–2016), multiplied by the age-adjusted all-cause mortality within 5 years (in lnHR) after the first diagnosis of the disease category.

### Disease weight
Disease burden of death, scaled 0–1, denoting the relative importance of a disease category.

### Disease incidence improvement
Age-adjusted hazard of being admitted to hospital for a disease category (excluding subsequent hospital visits for the same disease category) from one decade of birth to the next.

### Disease survival improvement
Age-adjusted hazard of dying within 5 years after the first hospital admission for a disease category in 2011 compared with having the first hospital admission in 2001.

### Disease improvements
Linear combination of the disease survival and disease incidence (averaged across decades of birth) in units of lnHR.

### Morbidity-driven mortality
The change in mortality rates expected from the improvements in morbidity (ie, weighted sum of disease survival and incidence for all 28 diseases).

All model coefficients used in the results can be found in online supplementary file 9.

## RESULTS
### Mortality

The population consisted of 1 967 130 Scottish individuals aged 35 years or older at the start of the study period (1 December 2000). About 53.3% were female, 78.5% had been admitted to hospital at least once within the study period and 30.6% died over the course of the study (31 January 2016). See table 1 for detailed population characteristics.

Quantifying mortality effects using Cox proportional hazard models, we observed statistically significant associations ($p < 1 \times 10^{-26}$) between mortality and sex, deprivation and decade of birth (online supplementary file 10). Women showed lower overall age-adjusted mortality rates compared with men (HR 0.71; 95% CI 0.70 to 0.71), corresponding to an expectation of life of 3.5 years longer than their male counterparts, while individuals from the most deprived areas (top decile) suffered mortality rates more than twice as severe (2.07; 95% CI 2.04 to 2.09) as those from the least deprived areas (bottom decile), corresponding to a difference in around 7.3 years of life. Median survival of men and women in the most deprived areas was 71.1 and 76.6 years, respectively, compared with 82.2 and 85.2 in the least deprived areas (online supplementary file 11). A wide gap between the most deprived decile and the adjacent one for men is apparent visually: the difference in median survival between deprivation deciles 1 to 9 is roughly constant (0.82/1.05 years per decile for women/men), but moving from the ninth to tenth deprivation decile has a greater effect, especially for men (1.99/2.67 years for women/men). Lastly, individuals born in the decade commencing 1935 had age-adjusted mortality rates 2.45 (95% CI 2.39 to 2.51) times of those born three decades later, corresponding to a difference in life expectancy of around 9 years of life.

### Morbidities and consequent mortality

Multiplying total number of hospitalisations during the study period (as a proxy for disease prevalence) by 5-year mortality after hospital admission (as a proxy for disease severity) provided a weight for the death burden of hospitalisation of each ICD-10 block. We restricted our analyses to 28 of the top disease blocks for burden of death (T-28, see Methods section). Among the T-28, total cases of disease incidence (ie, first-time admissions) during the study period ranged from 33 613 (A30–A49, 'Other bacterial disease') to 225 504 (R00–R09, 'Symptoms and signs involving the circulatory and respiratory systems'; table 2). Per-person total cases of disease incidence (not age-adjusted) were 68.0% higher for the most deprived decile (188 905 individuals with 331 701 first-time admissions) compared with the least deprived decile (187 193 individuals with 195 617 first-time admissions). Between sexes, per-person incidence was 2.2% higher for men (917 788 individuals with 1 257 417 first-time admissions) compared with women (1 049 342 individuals with 1 407 223 first-time admissions; online supplementary file 12).

In the first 5 years, the highest all-cause mortality rate was for patients admitted for C76–C80 ('Malignant neoplasms of ill defined, secondary and unspecified sites'; HR 26.1) compared with all-cause mortality rates for those not admitted for C76–C80. The lowest 5-year all-cause mortality rate was for those admitted for K20–K31 ('Diseases of oesophagus, stomach and duodenum'; HR 1.8) compared with mortality rates for those not admitted for K20–K31. Ordering diseases by their burden of death weights, we found 'Influenza and pneumonia' (J09–J18), 'Symptoms and signs involving circulatory and respiratory systems' (R00–R09) and 'Malignant neoplasm of respiratory and intrathoracic organs' (C30–C39) were the disease categories responsible for the most death (table 2), together accounting for 19% of the total death burden of the T-28 diseases.

Apart from sex-specific cancers, we observe significant differences in burden of death between men and women for injuries to the hip and thigh (S70–S79) and head (S00–S09), with the former having a higher burden in women due to more female cases and the latter having a higher burden in men due to more male cases. For both disease blocks, the effect of hospitalisation on subsequent mortality is greater in men than women (S70–S79 HR men: 3.19, women: 2.44; S00–S09 HR men: 2.32, women: 1.88). Strikingly, 5-year mortality after hospital admission for IHD is higher for women (HR 2.01/1.70 women/men), but this is offset by the lower prevalence of hospitalisation in women (online supplementary file 12).

### Trends in disease

To understand changes in disease survival rates, we next modelled the effects of a disease on all-cause mortality by year of hospital admission for admissions between 2001 and 2011 and 5-year survival subsequent to admission. We find an overall improvement over time in patient survival following hospitalisation, with a median decline between 2001 and 2011 in the 5-year HR of 16.8% for admitted cases across the T-28 diseases. The biggest improvements were for malignant neoplasms of the breast (C50) and male genital organs (C60–C63), which have seen 68.7% (95% CI 64.1% to 72.7%) and 50.2% (95% CI 42.9% to 56.6%) declines in the 5-year HR between 2001 and 2011 mortality, respectively. On the other hand, 'Other acute lower respiratory infections' (J20–J22) and 'Other diseases of the urinary system' (N30–N39) have seen increases in mortality hazard of 11.3% (95% CI 7.0% to 15.7%) and 24.0% (95% CI 14.6% to 34.1%), respectively (table 2; online supplementary files 13 and 14).

We next modelled age-adjusted incidence of hospitalisation for a disease by birth decade, under the simplified model that incidence is a cohort rather than period effect—essentially modelling that current incidence is the effect of (previous) lifetime exposures, rather than current exposures. We find disease incidence has fallen decade on decade of birth for cancers, cardiovascular and intestinal diseases, but this improvement appears to have slowed down in the last decade of birth (1955–1965)

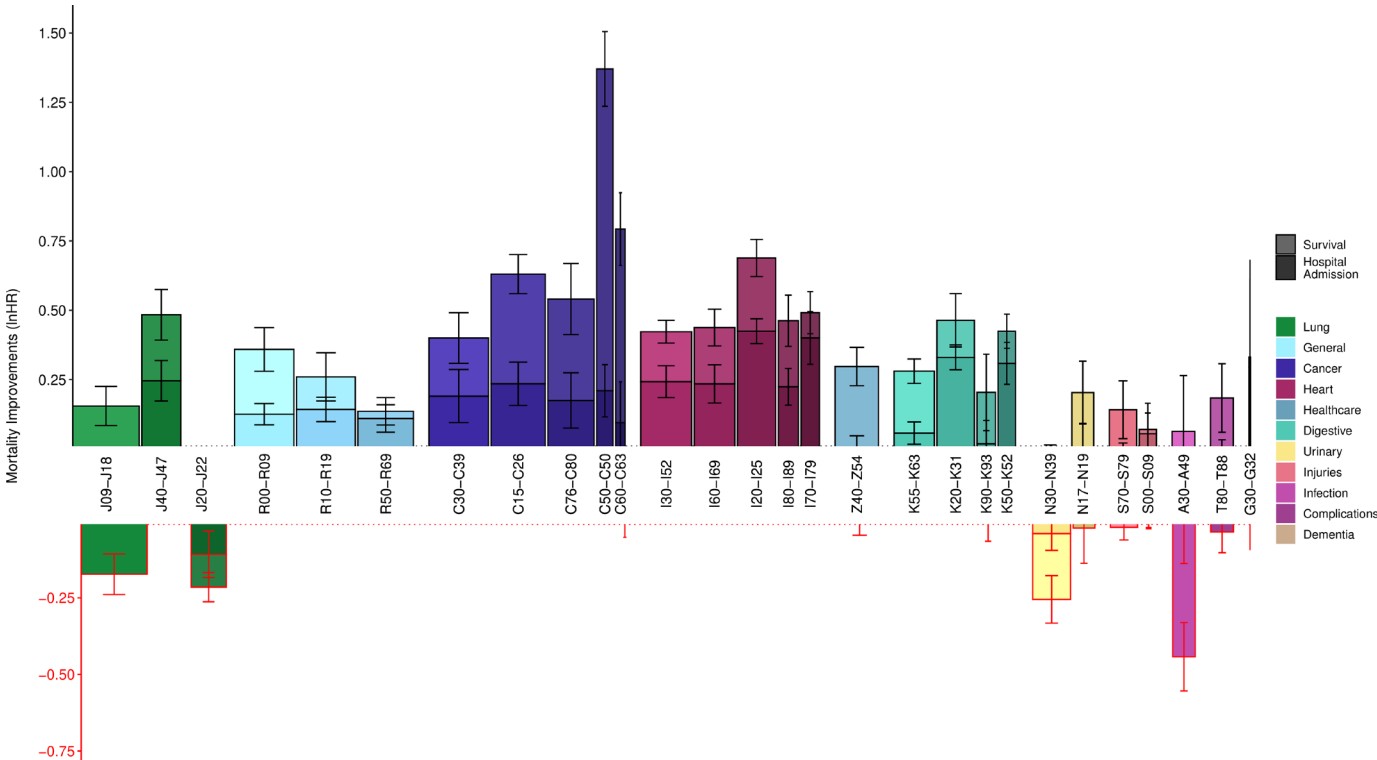

**Figure 1** Modelled decade-of-birth on previous decade-of-birth hospitalisations and survival show large improvements in cancer survival and heart disease incidence but deteriorations in infectious disease. Bars represent the mean improvements in hospital admission rate across decades of birth (darker bars), added to changes from 2001 to 2011 in 5-year survival rates following hospital admission (lighter bars). Both measures are expressed in age-adjusted terms. for definitions of each ICD-10 block, see table 2. Width of the bars represents the relative burden of death of each disease based on total first-time hospital admissions and 5-year mortality; as such, the total area of each bar represents the relative contribution to improvements—or deteriorations—in population mortality. Error bars are standard errors of the COX model coefficient. G30–G32 had too few hospital admissions to accurately model improvements (survival: lnHR 0.04, SE 0.14; hospital admission lnHR 0.29, SE 0.20). Z40–Z54 only showed improvements in survival.

considered. Age-adjusted incidence of 'Influenza and pneumonia' (J09–J18) and 'Other bacterial diseases' (A30–A49) has worsened by decade on decade of birth, over the whole range of births considered (online supplementary files 6 and 7).

When taking both trends in incidence and survival into account—adding (1) the average age-adjusted incidence rate reductions between decade on decade of birth to (2) the 2001–2011 reductions in 5-year disease mortality (online supplementary files 15 and 16)—we observe the death burden of cancers is declining most (figure 1). Notably, breast and prostate cancers have seen the largest improvement of all disease categories in the last decade. 'Other diseases of the urinary system' (N30–N39), 'Other bacterial diseases' (A30–A49), 'Other acute lower respiratory infections' (J20–J22) and 'Influenza and pneumonia' (J09–J18) have all seen increases in their effect on age-adjusted all-cause mortality.

Overall, we see broad consistency in the scale of improvements across decades of birth, except for 'Malignant neoplasms of respiratory and thoracic organs' (C30–C39), where we see greater decade-on-decade improvements among later decades (figure 2). Averaging these individual disease effects on death, using burden

of death weightings, we can then compare the modelled death rates with those observed, and see broad correspondence, with the 1935 and 1945 decades, showing the greatest improvements. Overall, our morbidity model suggests individuals from each successive decade of birth experience an average mortality rate of 0.74 (gaining ~3 years of life) compared with the previous decade of birth (table 2).

The shape of these disease-modelled mortality improvements by decade of birth broadly track the observed changes (figure 2). This is especially apparent when stratifying the improvements by sex: online supplementary file 17 shows a reasonable relationship between the projected morbidity driven mortality and observed mortality (ie, mortality trends in the study can largely be explained by trends in disease incidence and survival). Across sex and deprivation strata, taking into account disease survival improvements between 2001 and 2011 and all improvements in disease incidence between decades of birth, we find the largest reductions in death are due to improvements in 'Ischaemic heart diseases' (I20–I25), 'Malignant neoplasms of digestive organs' (C15–C26) and 'Malignant neoplasm of respiratory and intrathoracic organs' (C30–C39), while the largest increases in death are due

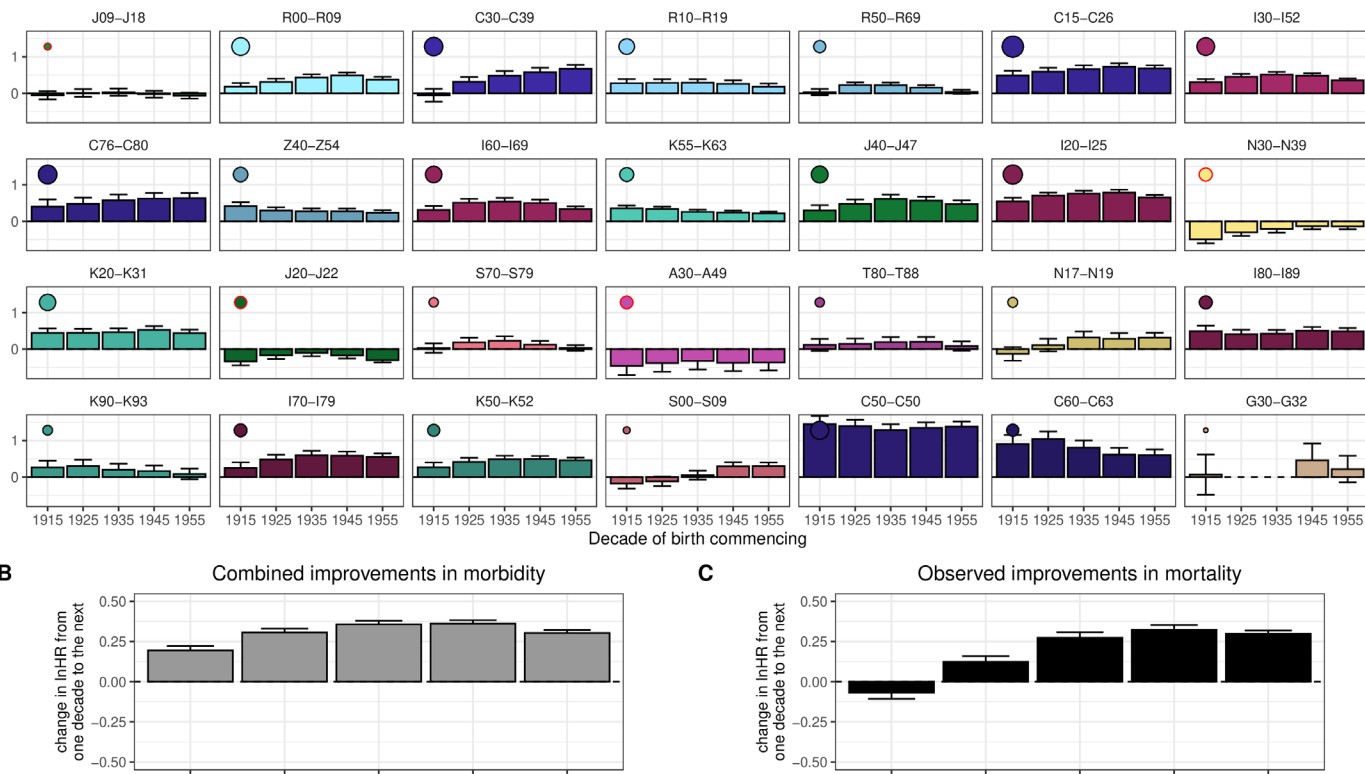

**Figure 2** Modelled decade-of-birth on previous decade-of-birth mortality reductions due to morbidity changes broadly track observed trends in mortality. Panels represent the combined improvements in hospital admission rate and 5-year mortality rates following hospital admission, expressed in age-adjusted lnHR and split by decade of birth under the model where change in incidence of disease is modelled by decade of birth and added to the survival effect is the change in subsequent 5-year survival rates from incidences in 2001 and 2011. (A) Improvements for each ICD-10 disease block (for definitions see table 2). Dots here represent the relative contribution of the disease to the overall improvements in morbidity-driven mortality, with larger dots indicating a greater contribution to morbidity improvements. A red circle around the dot indicates a negative contribution (ie, deterioration). (B) Modelled trend in deaths based on the weighted morbidities from the panels above. Diseases have been ordered by their burden of death (table 2), so smaller bars in early panels may have similar effect on the grey bar average (indicated by the dot size) as larger bars in later panels. (C) Observed trend in actual deaths from death records, by decade of birth, for comparison. see online supplementary file 18 for this graph stratified by sex and deprivation.

to 'Other bacterial diseases' (A30–A49) and Influenza and pneumonia' (J09–J18; online supplementary file 18). In addition, the deterioration in 'Other diseases of the urinary system' (N30–N39) morbidity shows a consistent increase with deprivation, while 'Other diseases of the digestive system' (K90–K93) shows consistently larger improvements in more deprived classes (online supplementary files 12 and 19).

Overall, we estimate 61.2% (95% CI 29.9% to 92.6%) of the improvement in mortality rates was due to improvements in survival following hospital admission, with the balance arising from reduced (age-adjusted) admission rates (table 2). Improved outcomes for cancers (C) were particularly driven by postadmission survival, especially C60–C63 (88% of mortality improvement attributable to survival rather than incidence), C50 (85%) and C76–C80 (68%), whereas for cardiovascular diseases (I) the balance was more even, as seen in I80–I89 (52%), I60–I69 (46%), I30–I32 (43%) and I20–I25 (38%).

As previously noted, disease severity was defined as the log HR for subsequent all-cause mortality among those with a previous admission for an index group of conditions compared with those with no such admission. We regarded the rate of improvement in disease severity over time as being constant if there was the same relative fall in log hazard rate over successive time periods (so for eg, we regarded a fall in lnHR from 0.6 to 0.3 as equivalent to a fall from 0.3 to 0.15). Assuming the improvements in survival following hospitalisation continue for the coming decade, and differences between incidence in birth cohorts remains the same, we project a 21% slowing of improvements in mortality (−0.242 lnHR, cf. −0.305 lnHR; table 3). Essentially, at least arithmetically, the population mortality benefits from improved cancer treatments in 2001–2011 will be hard to repeat as so much benefit has already accrued. Admittedly, this is a consequence of our model: essentially judging it equally difficult to reduce 50 excess deaths following cancer hospital

**Table 3** Mean (over birth decades) decade of birth on decade of birth improvements in morbidity for the study period, and projections into the subsequent decade by sex and deprivation

| Stratified | Group | Current improvements | | | Projected improvements | | |
|---|---|---|---|---|---|---|---|
| | | Hospital admission rate (lnHR) | Five-year mortality after admission (lnHR) | Combined (lnHR) | Hospital admission rate (lnHR) | Five-year mortality after admission (lnHR) | Combined (lnHR) |
| None | | −0.1182 | −0.1866 | −0.3047 | −0.1182 | −0.1235 | −0.2418 |
| Sex | M | −0.1428 | −0.1913 | −0.3340 | −0.1428 | −0.1273 | −0.2701 |
| Sex | F | −0.0971 | −0.1823 | −0.2794 | −0.0971 | −0.1130 | −0.2101 |
| SIMD | 1 | −0.1154 | −0.2204 | −0.3360 | −0.1154 | −0.1280 | −0.2433 |
| SIMD | 2 | −0.1157 | −0.1592 | −0.2743 | −0.1157 | −0.0456 | −0.1610 |
| SIMD | 3 | −0.1202 | −0.1186 | −0.2392 | −0.1202 | −0.0370 | −0.1572 |
| SIMD | 4 | −0.1418 | −0.1759 | −0.3201 | −0.1418 | −0.0541 | −0.1982 |
| SIMD | 5 | −0.1231 | −0.1731 | −0.2958 | −0.1231 | −0.0914 | −0.2145 |
| SIMD | 6 | −0.1305 | −0.1928 | −0.3227 | −0.1305 | −0.0998 | −0.2303 |
| SIMD | 7 | −0.1233 | −0.1787 | −0.3044 | −0.1233 | −0.1109 | −0.2343 |
| SIMD | 8 | −0.1305 | −0.2097 | −0.3402 | −0.1305 | −0.1422 | −0.2726 |
| SIMD | 9 | −0.0926 | −0.1619 | −0.2546 | −0.0926 | −0.1033 | −0.1959 |
| SIMD | 10 | −0.0950 | −0.1970 | −0.2920 | −0.0950 | −0.1092 | −0.2043 |

Mortality improvements were estimated from morbidity records by combining the mean improvement in hospitalisation rate across birth cohorts and the improvement in disease severity between 2001 and 2011. This was then projected forward assuming improvements in age-adjusted hospitalisation rate between birth cohorts remained constant and improvements in severity remained proportional to the (now reduced) overall mortality of the disease group.
SIMD, Scottish Index of Multiple Deprivation.

admission associated to 25, as it was to reduce from 100 to 50 and as such should be considered speculative. On the other hand, our model is clearly valid *in extremis*: if all excess cancer deaths were eliminated, no further cancer-driven improvement in mortality would be possible.

## DISCUSSION

In a study of 1 967 120 lives and 10 718 084 hospital admissions, we observed a median age at death of 82.2/85.2 for men/women in the highest socioeconomic decile, and 11.1/8.6 years less for the lowest decile. Cancers (C), cardiovascular disease (I), respiratory diseases (J) and unclassified symptoms and signs (R) were the principal ICD-10 chapters recurring in the T-28 disease blocks, where hospital admission was associated with the greatest subsequent all-cause mortality, which was a product of the rate of first hospital admission with group of conditions and of all-cause mortality in the 5 years following admission with that condition. Specifically, our top five causes of hospitalisations associated with subsequent burden of all-cause deaths were, in descending order, 'Influenza and pneumonia' (more common and with higher subsequent mortality than the average T-28 disease), 'Symptoms and signs involving the circulatory and respiratory systems' (common), 'Malignant neoplasm of respiratory and intrathoracic organs' (higher mortality), 'Symptoms and signs involving the digestive system and abdomen' (common) and 'General symptoms and signs' (common and higher mortality). While the latter might appear a benign diagnosis, our results suggest it is a fairly strong and frequent marker of subsequent all-cause mortality.

Across decades of birth, we modelled a reduction in mortality hazard of 0.737 (95% CI 0.730 to 0.745) due to improvements in morbidity, which broadly tracked improvements in observed mortality. The modelled improvement was 61% accounted for by reduction in excess mortality subsequent to admission and 39% accounted for by a fall in incidences of disease (as measured by hospital admission rates). The important (ie, burden of death weighted) improvements in incidence were driven by cancers and heart disease, while improvement in outcomes following admission were mostly driven by cancer, particularly breast and prostate cancer. In contrast, we found deteriorations in the incidence of bacterial disease and in mortality following admission for respiratory and urinary infections. Levels of morbidity and mortality varied strongly across socioeconomic groups, but patterns in *changes* of such were generally less apparent. Men showed greater rates of improvement in mortality and morbidity than women, with lung and throat cancers contributing most to male improvements and IHD contributing most to female improvements.

In conclusion, we find trends in morbidity appear to partly explain trends in mortality. The progress in prevention and cure within oncology and prevention of heart disease account for the greatest parts of mortality improvement in 2001–2016, and our model suggests mortality improvements may slow, simply because the

absolute effect of progress in treatment of these diseases will be difficult to repeat. However, there is scope for further improvements in life expectancy, especially if new progress is made in the treatment of other diseases associated with death, or if prevention initiatives accelerate.

## Strengths and weaknesses of the study

This study has avoided some of the known issues with cause of death recording[17] since it does not use cause-specific mortality and tracks wider disease effects and subsequent mortality (such as frailty) beyond direct causes of death, by combining hospitalisation and death records. Implicit tracking of underlying causes through an associated effect (admission to hospital for a disease) may improve estimates of trends in mortality, even if the underlying cause is obscure. We are also able to partition trends in deaths due to a disease based on trends in prevalence and incidence, which has been done for IHD,[18] but not simultaneously across diseases in the same dataset. Also, our results are unaffected by population shifts as we excluded immigrants into Scotland after 2001, and instead reflect trends within the defined groups. Combined with the scale of our data, this consistent tracking has enabled us to make like-for-like comparisons of the mortality outcomes of different disease classes across socioeconomic groups and their trends over time.

However, this study also has a number of limitations, relating to the population under study, the definition of diseases, considerations with hospital admission data, and modelling assumptions. Further discussion of these limitations can be found in online supplementary file 20.

In brief, we excluded migrants out of Scotland because their subsequent trajectory (especially death) could not be tracked. Migrants may be healthier than the average individual and excluding them could therefore overestimate the incidence of disease and death in the population we studied. However, our observed trends should remain unaffected if migration patterns did not change significantly during the study.

Second, for practical reasons we grouped the main diagnoses of hospital admissions by ICD-10 blocks and excluded any secondary diagnoses. As a result, we are not able to comment on the trends of individual diseases within blocks (which could offset each other) nor the trends or effects of comorbidities. The latter may affect our results if comorbidities have changed over time or by socioeconomic status; for example, a decline in lung cancer over time as a competing risk for heart disease would inflate the observed improvements for heart disease. However, this is likely to be partially mitigated by reductions in mortality for individuals not admitted for heart disease. Future work may account for comorbidities more explicitly by using competing risk regression and site-specific survival.

Thirdly, the first hospital admission on record and its date is only a proxy for incidence of severe disease. Excluding subsequent hospital admissions may understate the burden of diseases which have recurring episodes (such as influenza), although trends in these diseases are unlikely to be affected given our definitions remained constant. Conversely, diseases such as dementia and multiple sclerosis which are generally managed in the community are unlikely to result in an (immediate) admission to hospital and are therefore not captured accurately in our study. Examining trends in these chronic diseases through GP records was outside the scope of this study, but integration of our results with future work on GP records is likely to refine overall morbidity estimates.

Another consideration with hospital records is their indirect link to death. This relationship can be confounded on the one hand by other health risks and lifestyle factors, and on the other hand by coding inaccuracies and changes in admission policies and screening. Our stratified analysis by sex and socioeconomic status partially mitigates the former, and coding inaccuracies are unlikely to affect disease trends if these inaccuracies are stable over time. There is evidence that screening policies and hospital usage has changed during the study period, but their influence is limited and the opposite effects on disease incidence and survival will mostly offset each other when looking at the effect of morbidity (eg, influenza) (online supplementary file 20). However, some caution is needed when interpreting the exact split between improvements in disease incidence and survival.

Lastly, our model assumed 1) disease incidence is a function of year of birth, 2) survival after hospital admission is a function of year of incidence, and 3) these hazards are proportionate. The first two assumptions are a simplification, but necessary given year of birth and year of incidence are completely confounded for a given age at incidence. As to the third point, while disease status itself is not always strictly a proportional hazard, trends in incidence hazard ratios between birth decades and survival hazard ratios between years of hospital admission should still be captured appropriately (online supplementary file 20).

## Strengths and weaknesses in relation to other studies

There was a degree of correspondence in the principal burdens assessed here and a recent study by the Scottish Burden of Disease study (SBD).[19] This study used the same population and the same study period but assessed years of life lost (weighting young deaths more as opposed to our method which counted all deaths equally), included individuals younger than 35 years old and used different disease groupings. Their principal burdens were IHD (ranked 13th in our list of burdens), tracheal, bronchus and lung cancers (3rd), chronic obstructive pulmonary disease (12th), stroke (10th) and Alzheimer's disease (–). Aside from Alzheimer's disease, discussed below, much of the distinction appears to arise from our observation of an association between death and admissions with indistinct diagnosis (not considered a valid specific cause of death by SBD). In the case of influenza and pneumonia, differences arise due to our study identifying a marker of frailty as well as a direct cause of death, combined with SBD

grouping influenza and pneumonia under lower respiratory infections. A relative strength of our study stems from usage of incident morbidity (as marked by hospitalisation) in advance of death, based on recorded diagnosis at the time of hospital visit, thus tracking remote effects such as long-term frailty rather than cause of death (which has known limited accuracy, particularly at older ages[17]). However, the principal strength arises from the ability to distinguish trends in incidence of morbidity from trends in subsequent survival. On the other hand, a relative weakness is that we are reliant on hospital admission as a marker of incidence; therefore, diagnosed or latent (presumably milder) cases in the absence of admission are not visible to us, leading for example to significant discrepancy with SBD in the apparent relative burden of Alzheimer's disease, likely due to an understatement of its importance in our results.

The closing gap in mortality between the sexes and its widening across social classes observed in our study is consistent with recent findings from the Office of National Statistics, summarised by Torjesen,[20] which looked at socioeconomic deprivation in England and Wales. Similarly, a recent study of health inequality in England found rising levels of lifespan inequality across socioeconomic groupings arising from increasing inequalities across a broad span of causes of death.[21] These studies had the advantage of a larger sample size (~7.5 million deaths cf 600 000 in our study) and could therefore track trends in mortality and cause of death between stratified groups more accurately. However, Scotland's unique linkage of death records and electronic health records through eDRIS allowed us to directly model changes disease mortality at an individual level (avoiding issues with cause of death recordings and shifts in population demographics). Our study has the advantage of partly explaining these trends in mortality inequality through changes in disease incidence and survival: men experienced greater improvements in incidence of lung cancer and survival following heart disease hospitalisation compared with women, while more socially deprived individuals (men and women combined) suffered worse deteriorations in infectious disease, especially for the incidence and survival of hospitalisation for urinary tract infections. However, in contrast to Bennett et al,[21] we do not find a clear pattern in overall morbidity improvements across socioeconomic deciles in Scotland, and we do not observe a widening inequality in cancer, respiratory and Alzheimer's disease morbidity within our study population, although we are underpowered to detect the latter and our disease groupings were not identical.

Lastly, a recent study of coronary heart disease mortality in Scotland, using a sophisticated model to apportion improvement between prevention and treatment, found improvements for coronary heart disease between 2001 and 2010 were similar across social classes, and reported 33%–61% of these improvements could be attributed to advances in treatment.[18] Given the very different methods, although studying the same population, there is reasonable concordance with our own study: we find roughly equal improvements in heart disease across social classes and estimate 38% (95% CI 28% to 48%) of these improvements stem from increased survival after hospitalisation for ischaemic heart disease. Hotchkiss et al[18] are able to further partition improvements by uptake of primary and secondary prevention drugs and treatments. Such detailed analysis of specific diseases has been beyond the scope of our study.

## Implications for clinicians and policymakers

Much of the improvements in mortality observed in Scotland between 2001 and 2016 can be attributed to reductions in morbidity, as captured by hospital admissions. While this study examined mortality and morbidity in the Scottish population only, there is a substantial concordance in mortality trends across high-income countries,[7] as well as similarities in disease-related mortality trends between Scotland and the rest of the UK,[6] warranting similar studies to be performed in other high-income countries. It is a testament to healthcare services that the majority of mortality improvements appear to stem from advances in disease survival postadmission. Observed improvements in cancer incidence and survival—especially breast and prostate—coincide with a continued effort within Scotland,[22] the UK[23] and other high-income nations[24] to improve prevention and care of these diseases. However, the rapid advances in survival of both heart disease and cancer modelled by our study between 2001 and 2011 will be hard to continue to the same extent, as so much progress has already been made. At the same time, the observed deteriorations in infectious disease coincide with global increases in antimicrobial resistance[25] and emphasise the need to prioritise research in this area: infectious disease will become a larger contributor to mortality and may contribute to a widening of health inequalities between socioeconomic classes. If these current trends in morbidity continue, we expect morbidity-driven improvements in mortality to slow down. However, the life expectancy gap between Scotland and other high-income countries[26] suggests further mortality improvements are possible. The rate of this improvement will hinge on whether advances in all major diseases categories— especially infectious disease— can catch up with the progress we have recently seen on heart disease and cancer, and whether preventable deaths from external causes (such as suicide and drug-related deaths), which cannot be accurately tracked using hospital admissions, decrease rather than rise.

## FUNDING
The study was funded by the Lloyds Banking Group, for the creation, curation and dissemination of knowledge in the public interest, in particular to improve estimates of future population size and morbidity and mortality rates to facilitate healthcare and other government planning. All analyses stratified by socioeconomic deprivation (ie,

tables 2 and 3, online supplementary files 7, 9–12, 14, 16, 18 and 19) were confidentially re-analysed using ten socioeconomic groups specified by Lloyds, rather than the SIMD. No Lloyds employees were granted access to individual patient data. Data access was granted to University of Edinburgh researchers only and only through the national safe haven by the Public Benefit and Privacy Panel for Health and Social Care under application number 1617–0255/Joshi.

The study and its design were conceived by the last author. The funder reviewed the design and said that adding stratification by socioeconomic status was a key requirement for meaningful analysis and their funding was conditional on this, a request to which the authors readily agreed. The funder was kept informed of interim analyses and reviewed the draft manuscript, occasionally suggesting additional analyses or requesting clarifications. The funders were not permitted to and did not request the removal of any results.

The last author and authors affiliated with eDRIS and NHS Scotland were compensated by the University of Edinburgh using the grant from Lloyds Banking Group, inline with normal pricing for the work undertaken. PT was funded by the MRC Doctoral Training Programme (MR/N013166/1) and the University of Edinburgh College of Medicine and Veterinary Medicine. JFW was funded by the MRC QTL in health and disease. Apart from Lloyds Banking Group, the funders had no role in the study design, data collection, analysis and interpretation, or the decision to submit the work for publication.

**Acknowledgements** The authors would like to thank Lloyds for the funding and in particular Craig Butler and Stuart McDonald for their engagement with the project. The authors would also like to thank Steve Pavis and Doug Kidd for their guidance with the Public Benefit and Privacy Panel for Health and Social Care.

**Author contributions** PRHJT performed the formal analysis, software, visualisation, writing the original draft and review and editing. JJK helped in data curation and writing the original draft. JM helped in review and editing. IG helped in review and editing. JFW helped in supervision, review and editing. HC helped in review and editing. CMF helped in the conceptualisation of the study design and review and editing. PKJ performed the conceptualisation of the study design, funding acquisition, investigation, methodology, project administration, and supervision, and helped with the formal analysis, software, validation, writing the original draft, and review and editing.

**Funding** This study was funded by Lloyds Banking Group, PLC, Medical Research Council (HGU QTL in health and disease), MR/N013166/1, College of Medicine and Veterinary Medicine, University of Edinburgh.

**Competing interests** PKJ reports grants from Lloyds Banking Group, PLC, during the conduct of the study. This grant was awarded to the University of Edinburgh and used in part to pay for research costs of JJK and CMF. PKJ also reports shares in Lloyds Banking Group, PLC, as part of a diversified portfolio. The remaining authors declare no competing interests.

**Patient consent for publication** Not required.

**Ethics approval** This study was approved by the Public Benefit and Privacy Panel for Health and Social Care under application number 1617–0255/Joshi. As clinical records are provided without explicit patient consent, the panel requires the public benefit of the research to clearly outweigh any impact on individual patient privacy, and appropriate safeguards and security to be in place to protect patients. The panel granted access to deidentified patient data, accessible only through the National Safe Haven and only by University of Edinburgh researchers with data safeguarding qualifications. In addition, all results were reviewed by eDRIS before extraction from the safe haven to ensure no potentially identifiable information was made public.

**Provenance and peer review** Not commissioned; externally peer reviewed.

**Data availability statement** No data are available. Individual deidentified participant data cannot be shared. All model coefficients used in the results can be found in online supplementary file 9. Statistical code and technical details are available upon request from the corresponding author.

**ORCID iDs**
Paul R H J Timmers http://orcid.org/0000-0002-5197-1267
Peter K Joshi http://orcid.org/0000-0002-6361-5059

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
