## [Reviewer comments · BMJ Open]

ARTICLE DETAILS

TITLE (PROVISIONAL)	Trends in disease incidence and survival and their effect on mortality in Scotland: nationwide cohort study of linked hospital admission and death records 2001–2016
AUTHORS	Timmers, Paul; Kerssens, Joannes Joseph; Minton, Jon; Grant, Ian; Wilson, James F; Campbell, Harry; Fischbacher, Colin; Joshi, Peter

VERSION 1 – REVIEW

REVIEWER	David Roder University of South Australia, Australia
REVIEW RETURNED	04-Oct-2019

GENERAL COMMENTS	Comments for the authors regarding “Improvements in the incidence and survival of cancer and cardiovascular but not infectious.....2001-2016” This is a well-written and sophisticated study based on Scotland-wide linked hospital and death data. It is a “big picture” study that is necessarily reliant upon use of routine administrative data which are likely to have sub-optimal accuracy and inconsistencies in data quality over time. Confidence in the findings is strengthened by their plausibility, particularly when compared with corresponding findings in the scientific literature (e.g., as relating to differences in life expectancies and other outcomes by sex and socioeconomic status, and time trends). The study illustrates the utility of data linkage for achieving population-wide coverage. There were questions that came to mind where a response from the authors would be appreciated, namely: 1. Hospitalization was used as a proxy for incidence Notwithstanding the understandable rationale for using this proxy, and the plausibility of results, it would have been an expedience strategy given lack of ready access to high-quality incidence data. What is the likelihood that differences across the study population and over time were influenced by differences in hospital access or choices of service options? Has there been a concerted effort to limit the use of hospitals and to divert patients to less-expensive community-based alternative services? Could this have affected time trends and sociodemographic differences (e.g., by SES status)? Have waiting lists increased, acting as a disincentive to seeking hospital care or causing delay with lead-time effects. Has screening led to earlier treatments and overdiagnosis, introducing artificial effects on survival and hospital incidence? Could this have contributed, for example, to the results for breast and prostate cancer?
---

	2. Analysis of comorbidities This was excluded as being out of scope. This is reasonable. But what would be the likely effects of not accounting for comorbidity on sociodemographic differences in survival and time trends? Were sensitivity analyses undertaken using competing risk regression or site-specific survival? 3. The Cox proportional hazards model was used. Were any tests of proportionality undertaken to test underlying assumptions? 4. What was the rationale for investigating cohort rather than period effects? This is an interesting and well-conceived study, but responses to these questions would be appreciated.
--	---

REVIEWER	Yuling Hong Centers for Disease Control and Prevention United States
REVIEW RETURNED	14-Nov-2019

GENERAL COMMENTS	This is a population-based study with the purpose to identify the causes and future trends unpinning improvements in Scottish mortality and quantify the relative contributions of disease incidence and survival. The author concluded that morbidity improvements broadly explain observed improvements in mortality, with progress on prevention and treatment of heart disease and cancer contributing most. The gaps between men and women's health are closing, but those between socioeconomic groups are not. The strengths of this study is the ability to link historical individual death and electronic health record in the whole population, which makes direct modelling analysis possible. The manuscript is well written overall but can certainly be improved for more clarity in some parts of the manuscript. The general comments this reviewer has are:  1. According to the recent report of "Scotland's Population - The Registrar General's Annual Review of Demographic Trends", the life expectancy at birth in Scotland decreased between 2014-2016 and 2015-2017. Why you chose the year of 2016 as the last year of your study to assess "mortality improvement". 2. Hospital admission is often used as the proxy of disease incidence. In your study, you also extracted records from the CHI dataset that includes the GP database. It is not clear whether you identified any disease incidence cases from the CHI dataset. If not, why not. In addition, does the hospital admission database include emergency visits and hospital outpatient visits? If not, but such data are available in separate datasets, it would be good to include these data in estimating disease incidence as large proportion of patients who treated at the emergency department and hospital outpatient clinics would not be hospitalized. Therefore, you would not under estimate the incidence. You should at least discuss this in the manuscript. Here are a few specific comments:  1. Page 2, lines 29-31. The statement can be misleading. The proportion of deaths from influenza and pneumonia" and "Symptoms and signs involving circulatory and respiratory systems" as the first diagnosis in the death certificate is smaller. Not sure what you meant about "most death". Is it based on modeling analysis? In addition, I understand that "Symptoms and signs involving circulatory and respiratory systems" is an ICD
--

	disease-coding block. Do you see any issues to lump circulatory and respiratory systems in your particular study? The temporal trends in the number of hospital admissions for those with symptoms and signs involving the circulatory system and the respiratory system might be in opposite directions. 2. Page 10, line 219: those who are not familiar with this field may not understand why you calculate 5-year mortality but not 1- year or 10-year mortality. It would be helpful to add a sentence or two for clarification. 3. Page 13, table 2: are the groups of diseases organized in any particular order? Are they In alphabetic order according to the first letter of the ICD codes, diseases of human body systems, descending or ascending order of the number of hospital visits? It is not easy to follow. 4. Page 16, line 364: I do not see that cardiovascular diseases as a group have been defined. Might be good to add ICD codes (I00-I99 if that is the case) after “cardiovascular diseases here.
--	--

REVIEWER	Qingfeng Li Johns Hopkins University, USA
REVIEW RETURNED	23-Nov-2019

GENERAL COMMENTS	As requested, I am only assessing the statistical methods and analyses. There are two major steps in the statistical analyses. The first step is linking the records across data sources. The step is well implemented. A suggestion is to add sensitivity analyses regarding the potential biases due to out-migration and exclusions in data management. The second step is the statistical modeling. The uses of the Cox PH model are appropriate for the research questions. And the models were thoroughly validated.
--

REVIEWER	Rosie Cornish University of Bristol, UK
REVIEW RETURNED	03-Dec-2019

GENERAL COMMENTS	This is a large study using routine health datasets to examine mortality following hospital admissions. The statistical methods appear to be appropriate, but I found the paper quite difficult to follow in places and had to re-read some sections several times to remind myself what was being measured and how. I feel that there are ways in which both the methods and the results could be outlined more clearly to assist the reader. Methods  1. It would be useful to have a definition of the study population at the beginning of the methods section – i.e. all those born between 1905 and 1965 and living in Scotland in ... 2. It is stated that 1,477,796 death records were received from the National Registry of Scotland, of which 699,093 could be matched. So, half the deaths were not matched. What were the reasons / likely reasons for this? Were these people who were not in the study population? If so, it would be worth stating this explicitly. 3. In the methods section, under “Disease classification”, it is stated that ICD10 chapters have been used to define “diseases blocks” and that, to define incidence, only “the first admission of disease for each individual” was used. If I have understood
---

correctly, this was done at a code level (e.g. R07), so different “diseases” could be counted as the same thing using this method? It should be made explicit in this section exactly at what level a “disease” was defined.

4. Relating to this definition of incidence, I can see that this makes sense for many of the diseases. However, if someone was admitted with pneumonia (or any other acute disease) in 2001 and then again in 2011 (for example), this would not be the same episode. The authors have acknowledged in the discussion that their definition may not work as well for acute conditions. However, it would also be worth discussing the likely implication(s) of using this definition in terms of their results.

5. What was done in cases where multiple ICD codes were recorded for the same admission? Was the primary diagnosis used? This should be made clear and will obviously affect the interpretation of the results.

6. The authors looked at mortality post-admission for different diseases. What did they do when an individual was admitted, for example, for influenza one year then a different disease (or more than one) subsequently? Was the death attributed to the most recent admission? And, was that person therefore classified as censored for the earlier disease admissions or were competing risks taken into account?

7. When modelling mortality, the authors state that they fitted sex, decade of birth and deprivation. Was this also based on age in years – as described for the other Cox models?

8. The authors state that they fitted a “3rd order polynomial regression” to estimates of 5-year mortality from 2001 to 2011. More detail is needed here as it is not clear what was actually done.

9. As mentioned above, I found I had to re-read the methods section a few times to remind myself what a particular measure was (e.g. burden, weighted improvements). It might be helpful somewhere towards the end of the methods section to give a summary of all the outcome measures and how these were calculated.

10. The main aim of the paper appears to be to study changes (improvements) in mortality and incidence. However, the study population is individuals born in 1905-1965 and resident in Scotland for at least some of the period from 2001-2016. Thus, individuals born 1905-1915 are necessarily those who have survived until age 86-96, individuals born 1915-1925 are those who have survived until 76-86, and so on. This is likely to have resulted in selection bias. The authors should discuss the likely impact of this.

Results

1. The columns of Table 2 should be labelled more informatively. I appreciate that the explanations of the headings are given in the table subheading but the table would be easier to read if the labels were clearer. Relating to this, log hazard ratios are not particularly easy to “digest”. Also, the last column of Table 2 gives a ratio (of improvements in disease survival to incidence of hospital admission), but I can’t find this described in the methods section.

2. It was also difficult to understand some of the figures. It would be preferable to have these labelled more extensively rather than having explanations in a very long footnote/title.

	Methods/Results/Discussion 1. The following paragraph is included in the results section: “We find disease incidence has fallen decade on decade of birth for cancers, cardiovascular, and intestinal diseases, but this improvement appears to have slowed down in the last decade of birth (1955-1965) considered. Age-adjusted incidence of influenza and pneumonia (J09-J18) and other bacterial diseases (A30-A49) has worsened by decade on decade of birth, over the whole range of births considered (S Figure 4).” As mentioned above, I am concerned about the potential impact of selection bias. The authors should particularly discuss this in relation to the results like those above where they describe trends across of decades of birth / across time (the study period). For example, although age-adjusted, any comparison of e.g. 1915 to 1965 in terms of say incidence of a given disease will necessarily only be based on a comparison of older individuals. Do the authors think this selection may explain why projected improvements in mortality for those born in the earlier decades agrees quite poorly with actual improvements whereas the agreement is better for later decades? There are other results that may need to be interpreted in light of this. Minor comments  1. It is stated that the study was reported in line with the RECORD guidelines. However, I could not see the checklist. 2. It is usual to give hazard ratios as ratios rather than percentages. 3. I believe the term “historical” should be used rather than “historic” (e.g. Methods – Community Health Index dataset: “Records were extracted from the historic and current. . .”) 4. The figures presented in Supplementary Figure 1 – Source data 1 are quite strange. Why is a mean date needed when the median is given? And what are the units of measurement for the median? It appears to be year as a decimal. Wouldn't it be better to give month and year? And is the SD given in years? Why not give a range or IQR, then this could be in the same units of measurement as the median (month and year)? 5. The acronym NRS is used as a subheading then only written in full in the subsequent text. 6. The term “InHR” is used in the text but not defined. 7. It is not appropriate to give a correlation for actual vs projected figures (Supplementary Figure 6). The projected figures could systematically underestimate the actual figures but still have a very high correlation – this gives no indication on levels of agreement.
--	---

VERSION 1 – AUTHOR RESPONSE

Comments by David Roder:

DR 010

1. Hospitalization was used as a proxy for incidence

Notwithstanding the understandable rationale for using this proxy, and the plausibility of results, it would have been an expedience strategy given lack of ready access to high-quality incidence data. What is the likelihood that differences across the study population and over time were influenced by differences in hospital access or choices of service options? Has there been a concerted effort to limit the use of hospitals and to divert patients to less-expensive community-based alternative services? Could this have affected time trends and sociodemographic differences (e.g., by SES status)? Have

waiting lists increased, acting as a disincentive to seeking hospital care or causing delay with lead-time effects. Has screening led to earlier treatments and overdiagnosis, introducing artificial effects on survival and hospital incidence? Could this have contributed, for example, to the results for breast and prostate cancer?

We appreciate the reviewer's recognition for the rationale and near practical necessity, and accept their concerns regarding trends in hospital access, service options, and screening are valid. We have added these limitations and their expected effect on our results to the discussion of the main text at lines 546-550. We now also discuss exactly to what extent waiting times and overdiagnoses have changed in Scotland in Supplementary Note 1 and highlight breast and prostate cancer screening as examples.

DR 020

2. Analysis of comorbidities

This was excluded as being out of scope. This is reasonable. But what would be the likely effects of not accounting for comorbidity on sociodemographic differences in survival and time trends? Were sensitivity analyses undertaken using competing risk regression or site-specific survival?

Yes, we recognise excluding comorbidities raises potential issues and have extended the discussion in the main text to bring out the potentially complex effects it can have on our results (lines 524-529). Due to computational limitations within the Scottish National Safe Haven, performing the suggested analyses within a reasonable timeframe is impractical; therefore, we have highlighted them for future studies.

DR 030

3. The Cox proportional hazards model was used. Were any tests of proportionality undertaken to test underlying assumptions?

We did not test for proportionality, and expect in some cases the effects will not be strictly proportional. However, the measured trends between decades of birth and years of admission are not particularly susceptible to non-proportionality, provided the population shape doesn't alter too much. We have added a comment regarding non-proportionality in the Discussion in the main text at lines 555-558 and further elaborate our reasoning in Supplementary Note 1.

DR 040

4. What was the rationale for investigating cohort rather than period effects?

Sorry for not making this clearer in the first place. It was driven empirically by the observation of decade of birth-specific changes. We have clarified this in the "Design" section of the Methods at lines 152-156.

—

Comments by Yuling Hong:

YH 010

1. According to the recent report of "Scotland's Population - The Registrar General's Annual Review of Demographic Trends", the life expectancy at birth in Scotland decreased between 2014-2016 and 2015-2017. Why you chose the year of 2016 as the last year of your study to assess "mortality improvement".

We apologise for not making this clear in the text. The ethical approval we were granted at the time of study design was for the study period 2001–2016 only and the associated data capture reflected this. We have added this detail to the beginning of the Methods at line 93.

YH 020

2. Hospital admission is often used as the proxy of disease incidence. In your study, you also extracted records from the CHI dataset that includes the GP database. It is not clear whether you identified any disease incidence cases from the CHI dataset. If not, why not. In addition, does the hospital admission database include emergency visits and hospital outpatient visits? If not, but such data are available in separate datasets, it would be good to include these data in estimating disease incidence as large proportion of patients who treated at the emergency department and hospital outpatient clinics would not be hospitalized. Therefore, you would not underestimate the incidence. You should at least discuss this in the manuscript.

Yes, CHI captures community illness. However, our study design, ethical approval, and data capture only determined people's existence within the CHI database, to establish the study population, and not their CHI illnesses. We relied on SMR01 hospitalisation records (which includes acute inpatient and day cases) to capture disease. We acknowledge disease incidence may therefore be understated, although presumably severe incidence is captured. We have added these considerations to the Discussion in the main text at lines 537-539. Supplementary Note 1 elaborates on the reasons why examining GP records was beyond the scope of this study.

YH 031 / YH 032

1. Page 2, lines 29-31. The statement can be misleading. The proportion of deaths from influenza and pneumonia" and "Symptoms and signs involving circulatory and respiratory systems" as the first diagnosis in the death certificate is smaller. Not sure what you meant about "most death". Is it based on modeling analysis? In addition, I understand that "Symptoms and signs involving circulatory and respiratory systems" is an ICD disease-coding block. Do you see any issues to lump circulatory and respiratory systems in your particular study? The temporal trends in the number of hospital admissions for those with symptoms and signs involving the circulatory system and the respiratory system might be in opposite directions.

Our apologies for being unclear regarding the "most deaths" statement. This indeed reflects our modelling of disease importance, based on the total number of cases during the study and the increase in deaths following diagnosis, compared to those without the diagnosis. We have now clarified this in the text at lines 27-28. Regarding ICD disease-coding blocks, we acknowledge there may be sub-trends within blocks, however we wanted an objective way to group diseases and chose ICD blocks as its level of granularity, (i.e. number of categories) was tractable. We have added the possibility of trends offsetting each other in the Discussion of the main text at line 523, and further elaborate with the example of "Symptoms and signs involving circulatory and respiratory systems" in Supplementary Note 1.

YH 040

2. Page 10, line 219: those who are not familiar with this field may not understand why you calculate 5-year mortality but not 1- year or 10-year mortality. It would be helpful to add a sentence or two for clarification.

Yes, we should have made this clear, and recognise that all these periods are common. We observe

different rates of decay of excess mortality across blocks and chose five years as a compromise, trying to capture the great majority of the effect of the episode, but no more than the effect of the episode. We have added this reasoning in the “Design” section of the Methods at lines 143-147.

YH 050

3. Page 13, table 2: are the groups of diseases organized in any particular order? Are they In alphabetic order according to the first letter of the ICD codes, diseases of human body systems, descending or ascending order of the number of hospital visits? It is not easy to follow.

We apologise for being unclear. The diseases are ordered by the burden of death weights with the most common and deadly diseases at the top. This has now been clarified in the text at line 343.

YH 060

4. Page 16, line 364: I do not see that cardiovascular diseases as a group have been defined. Might be good to add ICD codes (I00-I99 if that is the case) after “cardiovascular diseases here.

Thank you for the suggestion. We have added the first letter of the ICD10 codes for cancer (C) and cardiovascular disease (I) to more clearly mark which disease blocks are included.

—

Comments by Qingfeng Li:

QL 010

A suggestion is to add sensitivity analyses regarding the potential biases due to out-migration and exclusions in data management.

Unfortunately, we cannot capture the key outcome for the exclusions (i.e. death). It would be an interesting, albeit ambitious study, to follow the people who have migrated out of Scotland. We have added these points to the Discussion in the main text at line 515.

—

Comments by Rosie Cornish:

RC 010

1. It would be useful to have a definition of the study population at the beginning of the methods section – i.e. all those born between 1905 and 1965 and living in Scotland in ...

We thank the reviewer for their suggestion. A description of the final sample has now been added to the beginning of the Methods. See lines 94-98.

RC 020

2. It is stated that 1,477,796 death records were received from the National Registry of Scotland, of which 699,093 could be matched. So, half the deaths were not matched. What were the reasons / likely reasons for this? Were these people who were not in the study population? If so, it would be worth stating this explicitly.

The majority of these individuals died before the study start and were therefore not captured in the CHI database. In line with your suggestion, we now state this on line 124.

RC 030

3. In the methods section, under "Disease classification", it is stated that ICD10 chapters have been used to define "diseases blocks" and that, to define incidence, only "the first admission of disease for each individual" was used. If I have understood correctly, this was done at a code level (e.g. R07), so different "diseases" could be counted as the same thing using this method? It should be made explicit in this section exactly at what level a "disease" was defined.

Our apologies. We use the first admission of a disease category for each individual, so diseases within a disease category are not counted multiple times. We have made this more explicit throughout the Methods at various points. For examples, see lines 135-137, 179, and 213.

RC 040

4. Relating to this definition of incidence, I can see that this makes sense for many of the diseases. However, if someone was admitted with pneumonia (or any other acute disease) in 2001 and then again in 2011 (for example), this would not be the same episode. The authors have acknowledged in the discussion that their definition may not work as well for acute conditions. However, it would also be worth discussing the likely implication(s) of using this definition in terms of their results.

Agreed. We believe our definition of incidence will have understated the burden for diseases with recurring episodes somewhat, but the effect on the trends seems likely to be less affected. At your suggestion, we have now added this to the Discussion at lines 532-533.

RC 050

5. What was done in cases where multiple ICD codes were recorded for the same admission? Was the primary diagnosis used? This should be made clear and will obviously affect the interpretation of the results.

Yes, this is a good point, we feel consistent with our approach of first incidence, and the fact that the first recorded disease is intended to be the admission trigger we analysed to primary diagnoses only, excluding any secondary diagnoses. We have made this more explicit in the Methods at line 131 and the Discussion at line 522.

RC 060

6. The authors looked at mortality post-admission for different diseases. What did they do when an individual was admitted, for example, for influenza one year then a different disease (or more than one) subsequently? Was the death attributed to the most recent admission? And, was that person therefore classified as censored for the earlier disease admissions or were competing risks taken into account?

Yes, this bears some clarification. Please see comments under DR 020.

RC 070

7. When modelling mortality, the authors state that they fitted sex, decade of birth and deprivation. Was this also based on age in years – as described for the other Cox models?

Yes, we have modelled age, sex, deprivation, and decade of birth. The baseline hazard uses age in years. The hazard ratios represent the effects of the other covariates. We have made this more explicit in the Method section at lines 201-203.

RC 080

8. The authors state that they fitted a “3rd order polynomial regression” to estimates of 5-year mortality from 2001 to 2011. More detail is needed here as it is not clear what was actually done.

We apologise for the lack of detail. We have expanded this section in the Methods to explain the rationale behind the analysis, and the exact calculations we performed (lines 235-246).

RC 090

9. As mentioned above, I found I had to re-read the methods section a few times to remind myself what a particular measure was (e.g. burden, weighted improvements). It might be helpful somewhere towards the end of the methods section to give a summary of all the outcome measures and how these were calculated.

We thank the reviewer for this suggestion and have added a “Summary of outcomes” section to the Methods for improved clarity. See lines 269-295.

RC 100

10. The main aim of the paper appears to be to study changes (improvements) in mortality and incidence. However, the study population is individuals born in 1905-1965 and resident in Scotland for at least some of the period from 2001-2016. Thus, individuals born 1905-1915 are necessarily those who have survived until age 86-96, individuals born 1915-1925 are those who have survived until 76-86, and so on. This is likely to have resulted in selection bias. The authors should discuss the likely impact of this.

Please see longer reviewer comment (RC 130) below on this issue.

RC 110

1. The columns of Table 2 should be labelled more informatively. I appreciate that the explanations of the headings are given in the table subheading but the table would be easier to read if the labels were clearer. Relating to this, log hazard ratios are not particularly easy to “digest”. Also, the last column of Table 2 gives a ratio (of improvements in disease survival to incidence of hospital admission), but I can’t find this described in the methods section.

Thank you for your suggestions. We have expanded the headings of Table 2 to include more detail and have converted log hazard ratios to hazard ratios and confidence intervals. We have also added the missing section in the Methods to describe how ratios were calculated. See lines 252-254.

RC 120

2. It was also difficult to understand some of the figures. It would be preferable to have these labelled more extensively rather than having explanations in a very long footnote/title.

We apologise for the lack of labelling. We have updated our most complex figures (Figure 2, S Figure 5, S Figure 6) to have more informative headers and axis labels.

RC 130

1. The following paragraph is included in the results section: "We find disease incidence has fallen decade on decade of birth for cancers, cardiovascular, and intestinal diseases, but this improvement appears to have slowed down in the last decade of birth (1955-1965) considered. Age-adjusted incidence of influenza and pneumonia (J09-J18) and other bacterial diseases (A30-A49) has worsened by decade on decade of birth, over the whole range of births considered (S Figure 4)." As mentioned above, I am concerned about the potential impact of selection bias. The authors should particularly discuss this in relation to the results like those above where they describe trends across of decades of birth / across time (the study period). For example, although age-adjusted, any comparison of e.g. 1915 to 1965 in terms of say incidence of a given disease will necessarily only be based on a comparison of older individuals. Do the authors think this selection may explain why projected improvements in mortality for those born in the earlier decades agrees quite poorly with actual improvements whereas the agreement is better for later decades? There are other results that may need to be interpreted in light of this.

This is a good point identifying some subtleties and we understand the concern but feel it is mitigated by the fact we have split birth periods into ten-year intervals but have studied a fifteen-year interval, as recognised by "age-adjusted" in the comment. Thus, for example people born 1914 (i.e. during 1905-1914 inclusive) have potentially contributed ages 87-102 to the study, whilst those born 1925 (i.e. during 1915-1924 inclusive) contributed 77-92 using age as the baseline hazard means that these people are compared (only) over broadly the 87-92 age range (note the actual ranges are slightly more complicated due to births not all occurring on the same day of the year). See Supplementary Note 1.

RC 140

1. It is stated that the study was reported in line with the RECORD guidelines. However, I could not see the checklist.

We apologise for this. The checklist was prepared, but not properly transferred. We have now enclosed a copy.

RC 150

2. It is usual to give hazard ratios as ratios rather than percentages.

Yes, we have amended this in lines 311, 313, and 321. However, in instances where we express a relative change in hazard ratio, we used the percentage as this is more intuitive.

RC 160

3. I believe the term "historical" should be used rather than "historic" (e.g. Methods – Community Health Index dataset: "Records were extracted from the historic and current...")

Yes, this has now been changed.

RC 170

4. The figures presented in Supplementary Figure 1 – Source data 1 are quite strange. Why is a mean date needed when the median is given? And what are the units of measurement for the median? It appears to be year as a decimal. Wouldn't it be better to give month and year? And is the SD given in years? Why not give a range or IQR, then this could be in the same units of measurement as the median (month and year)?

Yes, we did not use the most intuitive descriptive measures. In line with the suggestion, we now use medians and interquartile ranges, with units in years for dates, and deciles for socioeconomic deprivation. We have now also clarified this in the legend.

RC 180

5. The acronym NRS is used as a subheading then only written in full in the subsequent text.

This has now been changed.

RC 190

6. The term "lnHR" is used in the text but not defined.

We have now defined the first usage of lnHR as "loge hazard ratio" in the Methods.

RC 200

7. It is not appropriate to give a correlation for actual vs projected figures (Supplementary Figure 6). The projected figures could systematically underestimate the actual figures but still have a very high correlation – this gives no indication on levels of agreement.

Yes, correlation does not describe these issues. However, we were interested in examining whether groups which had stronger morbidity improvements actually had better mortality improvements. This is suggested by visual inspection of the figure: there is a reasonable relationship between the actual and projected figures, but we accept the reservations around correlation. We have amended this in the text around line 415.

VERSION 2 – REVIEW

REVIEWER	David Roder University of South Australia
REVIEW RETURNED	10-Feb-2020
GENERAL COMMENTS	Thank you for considering the matters i raised in the first review. I have no further comments.
REVIEWER	Rosie Cornish University of Bristol, UK
REVIEW RETURNED	17-Feb-2020
GENERAL COMMENTS	The authors have addressed my previous concerns. I have no further comments.